# Drug Repurposing in the COVID-19 Era: Insights from Case Studies Showing Pharmaceutical Peculiarities

**DOI:** 10.3390/pharmaceutics13030302

**Published:** 2021-02-25

**Authors:** Milo Gatti, Fabrizio De Ponti

**Affiliations:** Pharmacology Unit, Department of Medical and Surgical Sciences, Alma Mater Studiorum University of Bologna, 40126 Bologna, Italy; milo.gatti2@unibo.it

**Keywords:** drug repurposing, COVID-19, pharmaceutical formulations, hydroxychloroquine, remdesivir, lopinavir/ritonavir, heparin, phosphodiesterase inhibitors, dietary supplements, ozone

## Abstract

COVID-19 may lead to severe respiratory distress syndrome and high risk of death in some patients. So far (January 2021), only the antiviral remdesivir has been approved, although no significant benefits in terms of mortality and clinical improvement were recently reported. In a setting where effective and safe treatments for COVID-19 are urgently needed, drug repurposing may take advantage of the fact that the safety profile of an agent is already well known and allows rapid investigation of the efficacy of potential treatments, at lower costs and with reduced risk of failure. Furthermore, novel pharmaceutical formulations of older agents (e.g., aerosolized administration of chloroquine/hydroxychloroquine, remdesivir, heparin, pirfenidone) have been tested in order to increase pulmonary delivery and/or antiviral effects of potentially active drugs, thus overcoming pharmacokinetic issues. In our review, we will highlight the importance of the drug repurposing strategy in the context of COVID-19, including regulatory and ethical aspects, with a specific focus on novel pharmaceutical formulations and routes of administration.

## 1. Introduction

The Coronavirus disease 2019 (COVID-19) is a respiratory tract infection caused by severe acute respiratory syndrome coronavirus 2 (SARS-CoV-2), responsible for variable clinical manifestations ranging from an asymptomatic infection to severe respiratory distress syndrome, possibly leading to multi-organ failure and death. It was first identified in Wuhan in the Hubei province of China at the end of 2019 [1]. COVID-19 was stated as a public health emergency of international concern on 30 January 2020 and was subsequently listed as a global pandemic on 11 March 2020 by the World Health Organization (WHO) [2,3]. As of 11 January 2021, more than 84 million cases and 1.8 million deaths were reported in 220 countries [4].

Currently, apart from vaccines as preventive medicinal products, the European Medicines Agency (EMA) and the Food and Drug Administration (FDA) have recommended granting marketing authorization only to one antiviral agent, remdesivir [5,6], originally developed for the management of an Ebola outbreak in Africa [7]. Notably, the Italian Medicines Agency (AIFA) recently called for a reassessment of remdesivir for the management of COVID-19, by issuing potential use restrictions [8]. This stems from the evidence reported in the Solidarity trial [9], a multi-arm multi-stage adaptive randomized controlled trial promoted by the WHO (NCT04315948), investigating the efficacy and safety of treatments for COVID-19 in hospitalized adults, in which remdesivir showed no effect on hospitalized COVID-19, as indicated by overall mortality, initiation of ventilation, and duration of hospital stay [10]. While vaccines against SARS-CoV-2 are becoming available in the first months of 2021 [11], effective and safe treatments of COVID-19 are urgently needed to face the global emergency.

In this setting, drug repurposing represents a useful and feasible strategy, allowing rapid study of different treatments, both with lower costs and reduced risk of failure, given that the safety profiles of the agents are typically well-established. Drug repurposing or repositioning involves identifying new uses for approved or investigational drugs that are outside the scope of the original intended or approved medical uses [12,13]. Being that the development of new drugs is a lengthy process and consequently unfeasible to promptly face the global pandemic, drug repurposing gained a key role in the management of COVID-19 [14,15,16,17,18,19]. However, it is important to emphasize that this strategy also requires time, funding, and drug development knowledge in order to assess the efficacy and safety of a specific agent for a novel therapeutic indication. Although, smaller and shorter studies are usually adequate, compared to a new developed medication [12].

The report of the AIFA concerning the consumption of drugs during the COVID-19 outbreak [20] and the analysis of ClinicalTrials.gov represent excellent evidence of the clinical relevance of the drug repurposing strategy in the COVID-19 era. In Italy, in March and May 2020, a significant increase in the use of azithromycin, hydroxychloroquine, tocilizumab, and darunavir/cobicistat (all well-known agents used for different disorders) was reported, compared to the three previous months. Furthermore, up to 23 November 2020, 3987 trials had been registered on ClinicalTrials.gov concerning COVID-19, most of them investigating “old” agents deliberately repurposed for this novel infection. A summary of repurposed agents for COVID-19 management according to severity of the disease and their current place in therapy is provided in Figure 1.

From a pharmaceutical point of view, the drug repurposing strategy to face COVID-19 has offered a unique opportunity to devise novel research strategies. In this setting, the repurposing of a specific agent for a different therapeutic indication may give rise to a range of safety and efficacy challenges that could be addressed through a pharmaceutical formulation or drug delivery approach [21]. Consequently, both reformulating already approved drugs to overcome physicochemical, pharmacokinetic or safety issues (such as instability, solubility, poor membrane permeability, compromised bioavailability, pulmonary delivery, inadequate duration of action, or side effects), and developing alternative and more appropriate routes of administration (e.g., pulmonary or intranasal to maximize exposure in the primary infection site of SARS-CoV-2) represent specific pharmaceutical strategies aimed at improving the risk–benefit profile of the repurposed agents [21,22].

The aim of this narrative review is to discuss pharmaceutical aspects of significant case studies involving repurposed drugs widely used during the COVID-19 pandemic, with a specific focus on pulmonary drug delivery.

## 2. Approaches for Drug Repurposing: Focus on Artificial Intelligence

Although in the past drug repurposing was often the result of serendipitous observations [22,23], biological experimental and computational approaches are currently implemented in order to repurpose agents in a quicker and more efficient way [24,25]. Two basic principles underlie the different approaches for drug repurposing. First, a single drug may interact with multiple targets, thus leading to the search for novel pharmacodynamic properties of known agents. Second, several diseases may show common pathogenic pathways, thus leading to the identification of novel indications for the known targets [24].

A summary of possible approaches to drug repurposing directly applicable to COVID-19 are shown in Table 1. Notably, the large amount of big-data easily available in online repositories of knowledge and omics allows implementation of different computational approaches (e.g., in silico drug repurposing approach), identifying a list of approved agents directly testable in preclinical and clinical trials for COVID-19 according to predicted ability of affecting viral replication [23,24,26]. In this setting, the implementation of artificial intelligence (AI) and machine learning methods may contribute to drug repurposing and development for COVID-19, possibly becoming a cornerstone of the computational approach [26,27].

Several reports of direct application of AI and deep machine learning to COVID-19 exist [28,29], with baricitinib identified as potential candidate drug precisely through the implementation of an AI-based approach [30]. Additionally, the AI-based strategy may play a key role in the development of novel drug delivery systems in order to maximize therapeutic efficacy in COVID-19 [26].

## 3. Regulatory Aspects and Ethical Issues in Drug Repurposing for COVID-19: A Double-Edge Sword

Regulatory aspects are critical determinants for the implementation of repurposed agents, taking greater relevance in situations such as the COVID-19 pandemic, when, given the urgent clinical need, the regulatory agency should expedite their decision process without compromising accurate risk–benefit assessment [25,31].

Focusing only on EU registration, the main, legal basis for drug applications for drug repositioning can be found in the Directive 2001/83/EC. Particularly, the application process for repurposed drugs can be performed through three different procedures: centralized, decentralized, and mutual recognition [25,32], although the centralized procedure is the most used [33]. Two different pathways exist for drug repurposing in the EU: (1) the complete dossier pathway, which is provided in Article 8(3) of Directive 2001/83/EC, representing the standard application pathway and composed of general administrative information, complete (non-)clinical data based on the applicant’s tests and studies, or bibliographic literature substituting certain tests or studies; and (2) the well-established use application pathway, which is provided in Article 10a of Directive 2001/83/EC, implemented for drugs that have been used for at least 10 years in the EU, with recognized efficacy and an acceptable safety level. The applicant is not required to provide (non-)clinical data in the dossier but may use existing literature [32].

Usually, one or more clinical trials are required for the new indication except for exceptional situations, and 5–10 years are needed for the registration. During emergency situations such as the COVID-19 pandemic, both clinical research and regulatory decisions require a resolute acceleration. Particularly in the last months, the EMA implemented specific initiatives for acceleration of development support and evaluation procedures for COVID-19 treatments and vaccines [34,35] (Table 2). Furthermore, all the other major agencies (United Kingdom, Japan, and the US) issued specific regulations for accelerated approvals requiring less than the usual evidence. However, the risk that different agents may be licensed with less available information concerning safety and efficacy than would normally be acceptable may not be negligible. This concern may be exemplified by the case of remdesivir, which received Japanese authorization under the Exceptional Approval Pathway [36], Emergency Use Authorization in the USA [6], and in the United Kingdom access was granted under the Early Access to Medicine Scheme [37]. Conditional approval for remdesivir by the EMA for use to treat COVID-19 in adults and adolescents with pneumonia requiring supplemental oxygen was given under the new accelerated pathway reported in Table 2 [5]. However, the latest evidence reported that remdesivir showed no effect on hospitalized COVID-19, as indicated by overall mortality, initiation of ventilation, and duration of hospital stay [9,10], thus highlighting the potential concerns associated with accelerated approvals of repurposed agents.

Consequently, ethical issues may arise in clinical trials investigating repurposed agents for the treatment of COVID-19, particularly when drugs are tested or licensed with less scientific evidence than would have been desirable. Several potential pitfalls regarding both efficacy and safety issues may arise in this context when emergency approval of repurposed drugs is implemented. The extent of the COVID-19 pandemic and its health and social repercussions have given life to a so called “drug repurposing tsunami” [38], clearly witnessed from the fact that more than 500 drugs, biological medications, dietary supplements, and herbal preparations have been the subject of clinical trials in patients affected by COVID-19. Notably, most of these drugs have well-established indications based on high-level scientific evidence, such as hydroxychloroquine and tocilizumab for rheumatoid arthritis or lopinavir/ritonavir for HIV. On the other hand, usually only low-quality studies conducted in the early phase of the pandemic provided the basis for the large number of these drugs repurposed for use in COVID-19. Therefore, when drug repurposing is implemented in emergency situations, such as the COVID-19 pandemic, several concerns arise in terms of upholding the quality of the research, maintaining a high level of evidence, and preserving the integrity of clinical research ethics [39]. Specifically, the subjects enrolled in trials involving accelerated-approval drugs could be unrepresentative of real-world populations strictly requiring the investigated treatment (e.g., patients affected by severe COVID-19 and multi-organ failure). Additionally, if a repurposed drug approved through the accelerated regulatory pathway becomes the standard of care in the treatment of COVID-19, this could affect the trial design of other repurposed agents. Consequently, studies with greater sample sizes and longer timelines could be required in order to demonstrate statistically significant superiority or non-inferiority compared to only supportive treatment. An additional consideration is that endpoints (usually surrogate) would be not necessarily be chosen by objective criteria, given the emergency approval status of the comparator [40]. Furthermore, but not less important, the large demand for repurposed agents showing little evidence for the new indication may affect the fair allocation of resources, depriving patients of an effective treatment if they are taking the drug for its original and well-established indication, as in the case of hydroxychloroquine and rheumatic diseases [41]. The potential risk of adverse events should be closely considered when repurposed agents with no clear efficacy are used in COVID-19 patients, as reported with hydroxychloroquine [42]. The withdrawal of emergency use authorization for a specific repurposed agent given the lack of efficacy or due to safety issues could further dangerously reduce public confidence in the drug, especially when it is widely used for other indications [43]. In this way, drug repurposing in COVID-19 may pose different ethical “dilemmas”: On the one hand, drug repurposing may be a huge opportunity for discovering effective treatments for the pandemic; on the other hand, the limited efficacy of accelerated approved treatments, coupled with safety issues and the lack of fairness in resource allocation among patients, are serious concerns that may not be negligible.

## 4. Pharmaceutical Considerations for Pulmonary Drug Delivery: Aid to Improve Efficacy and Safety for Repurposing Agents in COVID-19

Although nearly all suggested treatments for COVID-19 are based on systemic administration of repurposed agents, the lung represents the primary injury site. SARS-CoV-2, by binding to the angiotensin-converting enzyme 2 (ACE2) receptor in the lungs, may infect alveolar epithelial cells, vascular endothelial cells, and macrophages, leading to extensive apoptosis and vascular leakage caused by rapid and extensive viral replication evolving into acute lung injury (ALI) and acute respiratory distress syndrome (ARDS) [44]. Consequently, pulmonary administration of repurposed drugs may be a useful and complementary strategy in improving both the efficacy and safety of a specific treatment. This represents a great opportunity for pharmaceutical research in COVID-19 through the development of novel drug delivery systems and pharmaceutical formulations potentially providing increased drug accumulation at the target site of the disease [26].

Although pharmaceutical aerosol formulations are typically more sophisticated and less efficient than those for conventional routes of administration [45], pulmonary drug administration could exhibit certain treatment advantages in a pulmonary disease such as COVID-19. In this setting, pharmaceutical formulations for inhalation could allow for: (a) direct delivery of high concentrations to the disease site; (b) rapid clinical response; (c) reduction of side-effects; (d) bypassing of first-pass metabolism; and (e) achievement of similar or superior therapeutic effect with the use of a lower dosage compared to oral or parenteral formulations [44,46,47,48].

Nebulizers, metered-dose inhalers, and dry powder inhalers represent the available inhalation devices for pulmonary drug administration. The physicochemical characteristics of the drug and its formulation guide the choice of a specific inhalation device, with liquid formulations being administered by nebulizers and metered-dose inhalers, and solid formulations by dry powder inhalers [44,49,50,51,52,53]. Notably, the extent of inhaled drug accumulation and the site of drug deposition within the airways closely depend on the size of generated aerosol particles. Smaller particles usually exhibit a greater total drug accumulation in the lungs, achieving a distal airway penetration compared with larger particles. Given that SARS-CoV-2 mainly affects the lung structures situated in the lower respiratory tract, particles smaller than 2 µm in diameter are the most suitable in driving inhaled agents for the treatment of COVID-19 [44,46]. Nebulized unfractionated heparin, given its anti-inflammatory, antiviral, anticoagulant, and mucolytic properties, has been evoked as repurposed agent in COVID-19, based on previous preclinical and clinical studies in ALI and ARDS [54]. Furthermore, several lines of evidence reported a better outcome in critically ill patients affected by COVID-19 treated both with unfractionated heparin and low-molecular weight heparin [55].

However, potential limitations of the pulmonary route of drug administration should be addressed—considering its greater suitability in mild or moderate cases, the limited benefit in severe respiratory forms characterized by multiorgan failure, and potential issues in terms of safety.

In addition to direct pulmonary delivery, different types of drug delivery vehicles have been proposed for repurposed agents in the COVID-19 pandemic, including microbubbles, extracellular vesicles, nanoparticle drug carriers, and liposomes [56]. Additionally, intranasal antiviral drug delivery may provide a further strategy for the management of COVID-19, especially concerning the prevention of disease transmission and treatment of nasal symptoms [57].

It is important to emphasize that there is an urgent need to take into account pharmaceutic and pharmacokinetic principles to guide clinical trials involving repurposed agents for COVID-19. The case of ivermectin is clear evidence of this concern. Specifically, ivermectin showed in vitro efficacy against SARS-CoV-2 at a concentration of 5 µM. However, several pharmacokinetic issues (including the highly protein bound) limited the direct applicability of this observation, given that with the use of the highest reported dose (1.7 mg/kg instead of approved dosage of 0.2 mg/kg), the maximum peak concentrations of ivermectin were well below the half maximal inhibitory concentration (IC50) for SARS-CoV-2, as well as lung exposure was inadequate [58,59]. Despite these pharmaceutic and pharmacokinetic considerations demonstrating the unfeasible repurposing of ivermectin for COVID-19, currently there are still 44 ongoing clinical trials recorded on ClinicalTrials.gov (search performed on 28 November 2020) involving ivermectin in this setting, of which only one intends to investigate the efficacy of an intranasal formulation in achieving adequate lung exposure.

Finally, the close relationship between pharmaceutical formulations and pharmacokinetic/pharmacodynamic (PK/PD) features is well represented by the case of lopinavir/ritonavir, an HIV drug combination used as a COVID-19 treatment especially in the first phase of pandemic, and further showing no benefit beyond standard care [60]. The lopinavir/ritonavir combination exhibits common side effects including diarrhea, nausea, and liver damage. From a pharmacokinetic point of view, it has a short half-life (approximately 4–6 h); thus, systemic concentrations show wide fluctuations between peak and trough (up to 8–10-fold). The development of a controlled-release formulation exhibiting zero-order release kinetics to maintain the minimum effective drug concentration could reduce the occurrence of the abovementioned adverse effects via the minimization of the peak-trough effect [61].

Consequently, the development of pharmaceutic formulations coupled with implementation of PK/PD principles may improve both efficacy (as showed for inhaled formulations) and safety (as seen for controlled-release formulations) of repurposed drugs in COVID-19 (Figure 2).

## 5. Drug Repurposing in COVID-19: Case Studies

In the last months, several repurposed agents have been suggested and investigated in clinical trials against COVID-19 [62,63]. Notably, novel pharmaceutical formulations have been hypothesized and developed for different repurposed drugs in order to improve efficacy (e.g., inhaled or intranasal formulations to enhance delivery and penetration in infection site) and safety (e.g., minimization of side effects through local and targeted administration) or to allow their administration and use in special populations (e.g., authorized medicaments manipulation to achieve formulations suitable for nonresponsive patients; Figure 2).

We may classify repurposed agents in four main categories: (a) “old” antivirals and non-antiviral agents showing certain inhibitory activity against SARS-CoV-2 (e.g., hydroxychloroquine, lopinavir/ritonavir, remdesivir, colchicine); (b) drugs originally developed and approved for severe chronic obstructive pulmonary disease (COPD) or idiopathic pulmonary fibrosis (e.g., phosphodiesterase-4 (PDE-4) inhibitors, pirfenidone); (c) dietary supplements, micronutrients, nutraceuticals, and herbal medicines repurposed as adjuvant therapies in COVID-19; and d) gas mixture (e.g., ozone auto-hemotherapy, nitric oxide). For each of these categories, we present significant case studies of different compounds showing relevant pharmaceutical peculiarities for clinical practice.

### 5.1. Novel Pharmaceutical Approaches for Old Drugs

Notably, a close relationship between the severity of COVID-19 clinical scenarios (from outpatient to intensive care unit admission) and the use of specific repurposed agents (antiviral or anti-inflammatory properties) may be identified (Figure 1). Antivirals are more appropriate for the early phase of infection, while anti-inflammatory/immunomodulatory agents should be reserved for the treatment of systemic hyperinflammation characterizing the late stages [64]. Consequently, the potential efficacy both of a certain route of administration and innovative pharmaceutical formulations is strictly dependent on the stage of the disease, given that pulmonary delivery of repurposed agents may be efficaciously implemented only in patients who maintain spontaneous breathing and adequate pulmonary function, becoming more challenging in severe cases of respiratory failure requiring mechanical ventilation.

#### 5.1.1. Hydroxychloroquine

Hydroxychloroquine was one of the most commonly used repurposed agents during the first phase of the COVID-19 pandemic, by virtue of its supposed ability in interfering with virus–endosome fusion and inhibiting the glycosylation of ACE-2 receptors, coupled with immunomodulatory and anti-inflammatory properties [62]. Although due to the lack of efficacy in both postexposure prophylaxis and treatment of mild/moderate COVID-19, coupled with safety concerns (mainly QT prolongation with occurrence of fatal arrhythmias) [9,65,66,67,68], hydroxychloroquine is currently no longer used as a repurposed drug; it maintains a certain “historical” relevance given its massive administration in the initial phase of the pandemic (as evidenced by the 263 registered studies on ClinicalTrials.gov). Furthermore, several noteworthy pharmaceutical formulations were suggested and developed to improve the efficacy and safety of hydroxychloroquine in COVID-19.

Particularly, four different pharmaceutical approaches leading to novel hydroxychloroquine formulations may be identified: (a) chirality switch to improve safety; (b) inhaled formulations to improve pulmonary delivery; (c) authorized manipulation to obtain currently unavailable liquid dosage forms; and (d) zinc combination to enhance antiviral activity.

The development of single enantiomers from old racemic drugs represents a variant of a repurposed strategy [69]. The strategy of chirality switch consists of the development and therapeutic use of a single enantiomer from a chiral drug that had previously been developed and approved as a racemate or as a mixture of diastereomers [69]. Notably, both hydroxychloroquine and chloroquine are chiral drugs administered as racemates (i.e., each as a 1:1 mixture of two paired enantiomers, namely (S)-(+) and (R)-(−)), and chiral switch to their (S)-(+)-enantiomers may represent an innovative pharmaceutical approach given the better safety profile with regard to retinopathy and detrimental cardiac effects (chloroquine and hydroxychloroquine off-target activity on cardiac functioning could display stereoselectivity) [69,70].

It is suggested that the hydroxychloroquine concentration required to effectively clear 100% of SARS-CoV-2 in vitro might not be achieved with the usual oral dosage (800 mg/day loading dose followed by 400 mg/day for four days), and a higher dose may increase the risk of side effects, including cardiotoxicity [71]. Furthermore, given the high tissue affinity of hydroxychloroquine, its lung penetration is limited also when larger oral doses are administered [72]. In this context, the development of an inhaled formulation may be useful in improving the efficacy and safety of hydroxychloroquine. Particularly, several inhaled hydroxychloroquine formulations at different development stages were proposed: an inhalable crystalline powder for aerosol delivery [72]; a simple aqueous solution delivered via nebulization, previously investigated in phase I and II trials to assess efficacy and safety in asthma at a dosage up to 20 mg daily (20-, 40-fold lower than the oral dosing schedule used for COVID-19 treatment) without occurrence of cardiac adverse events or significant electrocardiogram (ECG) changes [73,74], and a liposomal formulation administered by intratracheal instillation in a murine model. The latter showed higher lung exposure (approximately 30-fold) and longer half-life (2.5-fold) coupled with lower serum exposure in terms of peak concentration and area under the plasma concentration-time curve (AUC) compared to intravenous formulation [75].

Oral hydroxychloroquine formulations on the market only consist of tablets designed to be ingested; thus, no liquid dosage forms suitable for children or critical non-cooperative patients with nasogastric tube are currently available. Furthermore, tablets contain insoluble excipient, potentially causing clogging of the nasogastric tube. Consequently, an authorized manipulation of commercially available hydroxychloroquine has been suggested to fill this clinical gap, leading to the development of an oral solution/suspension of hydroxychloroquine obtained by industrial tablet trituration followed by dispersion of the powder in different aqueous vehicles [76].

Finally, hydroxychloroquine exhibits zinc ionophore features, specifically targeting extracellular zinc to intracellular lysosomes, thus interfering with RNA-dependent RNA polymerase activity and coronavirus replication. It has been suggested that the association with zinc supplementation may enhance the inhibitory activity of hydroxychloroquine on SARS-CoV-2 replication, with ten studies registered on ClinicalTrials.gov investigating this hypothesis [77]. Furthermore, hydroxychloroquine could also be easily combined with glycyrrhizic acid, a natural product isolated form the roots (*Glycyrrhizae Radix*) of the plants *Glycyrrhiza glabra* (typically cultivated in Europe, henceforth called European licorice) and *Glycyrrhiza uralensis* Fisch and *Glycyrrhiza inflata* Bat (used in the Chinese Pharmacopoeia) [78]. This novel pharmaceutic formulation enhances not only hydroxychloroquine delivery through its entrapment into multilamellar vesicles, but also provides a synergic effect due to the antiviral and anti-inflammatory activities of glycyrrhizic acid [78], as further detailed in Section 5.3.3.

#### 5.1.2. Remdesivir

Currently, remdesivir is the only medication approved by the EMA and the FDA for the treatment of COVID-19 [5,6], although no benefit in terms of overall mortality, initiation of ventilation, and duration of hospital stay has been recently reported [9,10]. Remdesivir is a prodrug, available only as an intravenous formulation, of a nucleotide analogue intracellularly metabolized to an analogue of adenosine triphosphate, inhibiting viral RNA polymerases, originally developed for the Ebola outbreak [7].

On the basis of physicochemical/PK properties and preclinical data, remdesivir is expected to have low tissue distribution and penetration, especially into the lung, achieving inadequate pulmonary concentrations compared to IC50 for SARS-CoV-2 when used at the current dosage (200 mg loading dose followed by 100 mg daily for 5–10 days) [79]. Furthermore, the occurrence of adverse events, including hepatotoxicity, precludes the administration of a higher dosage. Consequently, it has been suggested that direct pulmonary delivery may provide higher lung concentrations of remdesivir and its active metabolites in patients affected by COVID-19, potentially with the use of a 4–10-fold lower dose [79,80]. However, it should be noted that pulmonary delivery of remdesivir may be efficaciously implemented only in patients who maintain spontaneous breathing and adequate pulmonary function, resulting in more challenge in cases of severe respiratory failure requiring respiratory support.

Both nebulizer inhalation, using the current lyophilized formulation of remdesivir at dosage of 50 mg in acute settings, and dry powder inhalation as mid-term strategy, using a formulation containing 2.5 µm remdesivir crystalline form mixed with differently sized particles of lactose (15 to 200 µm), were hypothesized to overcome the limited tissue and pulmonary distribution [79].

Finally, two different strategies—namely, modification of prodrug compound or implementation of nanoformulations enhancing tissue targeting and efficacy—were suggested to increase tissue and pulmonary penetration of remdesivir [79]. Currently, two phase 1 ongoing studies (NCT04480333 and NCT04539262) are evaluating the PK and safety of inhaled formulations of remdesivir.

#### 5.1.3. Lopinavir/Ritonavir

Lopinavir/ritonavir association was largely administered in the first phase of the COVID-19 pandemic, based on inhibitor activity on SARS-CoV-2 reported in in vitro studies, although clinical trials demonstrated limited efficacy [60,62]. However, to overcome different concerns associated with lopinavir/ritonavir in terms of safety and to promptly make available liquid formulations for oral administration, pharmaceutical research played a key role during the first weeks of the outbreak.

Given that lopinavir/ritonavir is marketed in Italy only in tablets, the available oral liquid formulation may be imported from other European countries [81]. However, during the first phase of the COVID-19 pandemic, the wide use of lopinavir/ritonavir in critical care settings for nonresponsive patients induced a shortage of the oral solution available in the European market [82]; thus, the implementation of a magistral formula preparation by the hospital pharmacy as a galenic oral formulation from the tablets represented a useful alternative for patients who required the administration of a lopinavir/ritonavir oral solution (e.g., children, critical care patients via nasogastric tube) [81,82].

As regards safety issues, the lopinavir/ritonavir combination exhibits several well-known adverse events (e.g., hepatotoxicity, metabolic disorders) coupled with extensive potential for causing relevant drug–drug interactions; thus, toxicity reduction and by-pass metabolic pathways would be desirable. Consequently, novel formulations of lopinavir/ritonavir based on the use of endogenous extracellular vesicles represent a suggested repurposing approach to improve safety and efficacy for protease inhibitors, providing a target and personalized therapy [83]. The development of extracellular vesicles represents a pharmaceutical approach previously supported for drug delivery in cancers and central nervous system disorders [84]. For COVID-19, extracellular vesicles may be isolated from the plasma of the patient, loaded with the lopinavir/ritonavir combination, and ultimately intravenously administered back to the same patients, thus being capable of delivering therapeutic agents to target tissues with few or no side effects [83].

Finally, as previously mentioned, the development of a controlled-release formulation, exhibiting zero-order release kinetics to maintain the minimum effective drug concentration, could improve safety profile of lopinavir/ritonavir via the minimization of the peak-trough effect [61].

#### 5.1.4. Heparin

Infection, inflammation, and coagulopathy caused by SARS-CoV-2 infection lead to endothelial dysfunction and microvascular thrombosis, responsible for worsening of pulmonary function and development of ALI and ARDS in critically ill patients [54,85]. The use of heparin, both in unfractionated or low-molecular weight formulations, was demonstrated to improve outcome and reduce mortality rate in COVID-19 patients [86]. Nebulized unfractionated heparin exhibited antiviral, anti-inflammatory, anticoagulant, and mucolytic effects with the benefit of reducing bleeding compared to systemic administration [54]. Nebulized heparin formulations demonstrated some benefits in preclinical and clinical non-COVID studies [54], and currently it is under investigation in four trials for the management of COVID-19 patients at a dose of 25,000 IU every 6 h for 10–21 days.

However, heparin is not effective in clearing fibrin clusters deposited in the alveolar space in patients affected by ARDS (including COVID-19 subjects); thus, fibrinolytic agents may play an important role in this setting [85]. Several case reports demonstrated that the administration of systemic tissue plasminogen activator (tPA) was associated with improvement in the respiratory status of patients affected by severe COVID-19 [87], although the risk of severe bleeding represents a serious concern.

In order to overcome this issue, a pharmaceutical formulation of nebulized tPA (alteplase) was developed and used with benefits in different cases of ARDS and plastic bronchitis [88]. Furthermore, an ongoing phase 2 study is currently assessing nebulized tPA for the treatment of plastic bronchitis. Consequently, it is suggested that this formulation of nebulized alteplase, coupled with the administration of inhaled heparin, may be useful for the treatment of severe COVID-19 patients, allowing for a more targeted approach while limiting the occurrence of severe bleeding [85]. An ongoing phase 2 study is currently assessing a formulation of nebulized rt-PA in COVID-19 patients requiring mechanical ventilation (NCT04356833).

A summary of relevant pharmaceutical approaches applied to “old” repurposed agents is provided in Table 3.

#### 5.1.5. Plitidepsin

Plitidepsin is a marine-derived anticancer compound isolated from the Mediterranean tunicate *Aplidium albicans*. It exhibits pleiotropic effects on cancer cells by binding to the host eEF1A2 protein, resulting in cell-cycle arrest, growth inhibition, and induction of apoptosis. Plitidepsin could be effective in patients with relapsed/refractory multiple myeloma, as reported in a phase 3 trial [89]. Recently, eEF1A2 protein has been described as an important host factor for the replications of SARS-CoV-2 [90]. Notably, both in vitro and in vivo preclinical studies reported that plitidepsin exhibited potent antiviral activity against SARS-CoV-2 (approximately 27.5-fold higher than remdesivir), although cytotoxicity was remarkable [90]. A recent phase 1/2 study found a significant reduction in viral load (up to 70% at 15 days) in hospitalized patients treated for three days with plitidepsin [90].

### 5.2. Agents Originally Developed and Approved for the Treatment of COPD and Idiopathic Pulmonary Fibrosis: The Role of Phosphodiesterase Inhibitors and Pirfenidone

#### 5.2.1. Phosphodiesterase-4 (PDE4) Inhibitors

Phosphodiesterase-4 (PDE4) inhibitors have been advocated as potential repurposed agents for the treatment of early phase COVID-19 patients or as prophylaxis for the elderly admitted to resident care facilities [91,92,93], given their multiple anti-inflammatory activities (including inhibition of interleukin-17, closely involved in ALI caused by COVID-19 [94]) coupled with a well-known excellent safety profile [92,93]. Both apremilast and roflumilast, respectively, approved for the management of severe psoriasis and COPD, are potential agents for repurposing in the management of COVID-19. Currently, three studies investigating apremilast for this novel indication are ongoing, while in a case report, beneficial effects were found in a patient receiving apremilast for severe psoriasis and affected by SARS-CoV-2 infection [95].

Additionally, pentoxifylline, a phosphodiesterase inhibitor showing anti-inflammatory activity, was suggested for repurposing in COVID-19, given its better safety profile compared to conventional immunosuppressors [96]. Currently, two ongoing trials are investigating the efficacy of pentoxifylline in COVID-19. Furthermore, a pentoxifylline dry powder formulation suitable for inhalation administration was developed for the treatment of smoke inhalation in burn victims [97]. This inhaled formulation was found feasible, and the addition of leucine improved the fine particle fraction of pentoxifylline, maintaining particle size in the optimal range (≤5 micron) [97].

#### 5.2.2. Pirfenidone

Relevant lung fibrotic evolution following COVID-19 has been identified, leading to severe clinical consequences responsible for impairment of respiratory function. Therefore, the use of antifibrotic agents was also suggested in the management of COVID-19, although their role is still poorly defined [98,99].

Pirfenidone is a synthetic compound marketed for the treatment of idiopathic pulmonary fibrosis, showing pleiotropic properties in terms of antifibrotic, anti-inflammatory, and antioxidant effects by inhibiting the proliferation of fibroblasts and the production of proinflammatory and profibrotic cytokines [98]. Pirfenidone displayed efficacy in diseases with profibrotic pathways activated by immune/inflammatory dysregulation showing certain similarities with SARS-CoV-2 infection [96], and an in silico model also identified its promising role for repurposing in COVID-19 [100]. Currently, an ongoing trial investigating pirfenidone in patients experiencing fibrotic lung lesions due to COVID-19 is registered.

However, oral pirfenidone exhibits substantial side effects (including hepatotoxicity), often limiting its use and effectiveness due to specific PK/PD properties (namely, large biodistribution and low potency) [101]. Consequently, large oral doses are required to achieve adequate lung concentrations, although the approved oral dose has been established near the upper safety threshold, and a further dose escalation is not allowed for safety reasons [101].

To overcome these concerns, a nebulized inhalation formulation was developed to maximize pirfenidone lung exposure [101,102]. This formulation was investigated in preclinical and phase 1 clinical trials, demonstrating that lung exposure was well above the pirfenidone IC50, with more than 15-fold lower systemic concentrations compared to the oral formulation [101,102].

### 5.3. Dietary Supplements, Micronutrients, and Herbal Medicines

Several dietary supplements, micronutrients, nutraceuticals, probiotics, and herbal medicines formulations were suggested as repurposed compounds for the management of COVID-19 [103,104,105,106,107,108,109,110,111]. Although evidence of their benefit in viral infections (e.g., influenza, common cold, or SARS) is limited [103], different ongoing trials are currently investigating the efficacy of several dietary supplements and herbal compounds in COVID-19 when used alone or in combination with “traditional” repurposed agents. Notably, while medicinal products are well defined for specific indications and they must follow specific legislation to demonstrate quality, efficacy, and safety to obtain marketing authorization, dietary supplements, nutraceuticals, and herbal products do not follow a similar procedure, given that marketing authorization is not required. Thus, their efficacy remains poorly supported and open questions sometimes remain on safety [112].

As regards Chinese herbal medicine, different meta-analyses of randomized controlled trials [113,114] reported a better outcome in patients treated with Chinese herbal medicine in association with traditional Western medicine compared to traditional Western medicine alone, although significant biases in included studies existed. Herbal formulations exhibit a wide range of pharmacological functions, including anti-inflammatory, antiviral, antipyretic, expectorant, anti-asthmatic, and antitussive effects [104,113,115]. Licorice root (Gancao, Radix Glycyrrhizae) was the most administered compound [113]. Different formulations of Chinese herbal products exist, including decoction, granule, capsule, oral liquid, pill, and injection, with decoction being the most commonly used [113]. However, it is noteworthy that more evidence in terms not only of efficacy but also, above all, in terms of safety are required for these compounds [116].

We focused on novel pharmaceutical formulations of dietary supplements and herbal medicines (namely zinc, essential oils, and glycyrrhizin) developed and implemented in COVID-19 setting.

#### 5.3.1. Zinc Supplementation

Zinc exhibits a wide variety of direct and indirect antiviral activities against different species, including rhinovirus and influenza virus, enhancing both immune and adaptive immunity, as well as affecting virus attachment and replication [117].

Although the efficacy of zinc supplementation in treating the common cold caused by rhinoviruses is debated [118,119], it is important to underline that intranasal zinc gluconate gel formulations exist, potentially providing for direct micronutrient delivery at the site of infection. The administration of intranasal zinc formulations could also be repurposed as adjuvant treatment in patients affected by COVID-19, particularly concerning the prevention of disease transmission and the treatment of nasal symptoms [57]. However, zinc toxicity involving the olfactory system was found in preclinical models, and several cases of zinc-induced anosmia syndrome were reported [120].

#### 5.3.2. Essential Oils

Essential oils (EOs) include a complex mixture of volatile phytochemicals from diverse classes, including monoterpenes, sesquiterpenes, and phenylpropanoids, showing anti-inflammatory, immunomodulatory, bronchodilatory, and antiviral properties [111,121,122]. EOs usually contain about 20–60 components showing widely different concentrations, of which two or three are present at higher concentrations (20–70%; major components) compared to the others (retrieved in trace amounts), thus determining the biological properties of the compound [123]. The chemical profiles of the EOs differ not only in the number and type of molecules but also in their stereochemical structures and can be very different according to the selected method of extraction [123].

Notably, EOs exhibited in vitro activity against several viruses, including influenza and other respiratory viral infections [121,122]. Different EOs have been investigated through different repurposing approaches (including the in silico approach, in vitro assays, molecular docking) against COVID-19, being eucalypt oil (*Eucalyptus globulus*, jensenone and eucalyptol as major components) and garlic oil, coupled with several single major components (namely farnesol, anethole, cinnamaldehyde, carvacrol, geraniol, cinnamyl acetate, l-4-terpineol, thymol, pulegone, eugenol, menthol, and carvacrol) the most promising compounds [121,122].

EOs are usually used by external application (gargles or inhalation), with the respiratory tract exhibiting the most rapid route of administration, followed by the dermal pathway [123]. However, unfavorable chemical properties of EOs—namely, poor solubility, solvent toxicity, high volatility, low bioavailability, and physicochemical instability (responsible for degradation of EOs components)—limited their use as active compounds in several formulations [123,124,125]. Consequently, the search for different novel formulations arose as an urgent need for pharmaceutical research in this area, also in order to potentially improve and implement the use of EOs in viral infections, including COVID-19, leading to the development of many nanotechnology-based carriers; namely, liposomes, dendrimers, nanoparticles, nanoemulsion, and microemulsion [123,124,125]. Encapsulation of bioactive and major compounds of EOs represents a feasible approach to modulate drug release, increase the physical stability of the active substances, protect them from interactions with the environment, decrease their volatility, enhance their bioactivity, reduce toxicity, and improve patient compliance and convenience [123,124,125].

Currently, there are different ongoing trials investigating EOs in the management of COVID-19, including their efficacy for anosmia recovery in post-COVID infection.

#### 5.3.3. Glycyrrhizin

Glycyrrhizin or glycyrrhizic acid, is a natural product isolated form the roots (*Glycyrrhizae Radix*) of the plants *Glycyrrhiza glabra* (typically cultivated in Europe, henceforth called European licorice) and *G. uralensis* Fisch and *G. inflata* Bat (used in the Chinese Pharmacopoeia). It has been demonstrated to exhibit antiviral (based on cytoplasmatic and membrane effects) and anti-inflammatory/immunomodulatory properties (through the activation of multiple pathways involving Toll-like receptors and inhibition of proinflammatory cytokines), inhibiting in vitro isolates of SARS-associated coronavirus and other respiratory viruses [78,104].

However, the extensive first-pass metabolism strongly reduces plasmatic exposure of glycyrrhizic acid, with the consequent achievement of inadequate serum concentrations, well below the IC50 for SARS-CoV [78]. To overcome this issue, different pharmaceutical approaches have been explored: (a) the modification of chemical structure of glycyrrhizic acid, in order to develop amide derivatives and amino-acid conjugates that may considerably enhance the activity against SARS-CoV but with increased cytotoxicity and (b) the development of drug delivery systems consisting of the encapsulation of glycyrrhizic acid into nano-liposomes, hyalurosomes, or niosomes [78]. The latter formulations not only may improve plasmatic bioavailability and exposure of glycyrrhizic acid but also facilitate the transportation and delivery of co-transported drugs, given the amphiphilic properties of vesicles, allowing for the enhancement of poorly soluble drugs and increasing the passive diffusion through cellular membranes of co-transported agents [78].

As previously mentioned, hydroxychloroquine, a repurposed drug widely used in the first phase of the COVID-19 pandemic, is one of the compounds combined with glycyrrhizic acid in novel nano-formulation by virtue of the potential antiviral and anti-inflammatory synergism coupled with an improvement in delivery [78].

### 5.4. Ozone

Ozone represents the triatomic form of oxygen, showing in vitro and in vivo antioxidant and anti-inflammatory activity when administered as a gas mixture of oxygen/ozone, obtained from the modification of medical-grade oxygen using an ozone generator device [126,127,128]. The use of ozone for the management of different orthopedic diseases, diabetic foot, and cutaneous infections caused by viruses or fungi is well-established [129]. Furthermore, the efficacy and safety of ozone administration in patients affected by HIV, hepatitis C, Ebola, and influenza was reported in several cases [130]. Consequently, by virtue of its antiviral properties, ozone has been also proposed for the management of COVID-19, showing promising activity in several case reports when administered as major autohemotherapy (MAH) for seven days [130].

The main antioxidant mechanism of an oxygen/ozone mixture relies on the concept of oxidative preconditioning, consisting of the modulation of the endogenous antioxidant system triggered by a calibrated oxidant stimulus [121]. Ozone treatment may be able to promote an adaptation to oxidative stress, improve tissue oxygenation, and induce anti-inflammatory cytokines while modulating anti-inflammatory mediators, thus explaining its potential antiviral activity [126,131].

The recommended systemic administration routes are ozonized saline solution, MAH, and extracorporeal blood oxygenation–ozonation (EBOO). Furthermore, a variant of the minor autohemotherapy was shown to easily induce the oxidation of free viral components using ozone concentrations near 90 µg/mL per mL of blood, thus theoretically representing an inactivated and immunogenic vaccine [126].

Currently, eight ongoing clinical trials are investigating the efficacy of ozone in COVID-19. Notably, in one of these (NCT04651387), a novel lipidic pharmaceutical formulation is being assessed to overcome the limited efficacy and feasibility of current ozone formulations, characterized by high volatility and requiring administration by auto-hemotransfusion. This novel formulation consists of the development of ozonized oil, characterized by high stability and bioavailability due to its binding with a lipid carrier. Additionally, ozonized oil administration is totally noninvasive, occurring by oral administration of pills or as a nasal spray, thus becoming also suitable for healthy subjects as potential prophylactic treatment.

## 6. Conclusions

COVID-19 has emerged as a very serious threat to global health. Unfortunately, no agents show clinical efficacy against SARC-CoV-2 and its complications, including remdesivir, the only drug currently approved for this indication, thus dampening the initial excitement for the drug repurposing strategy.

However, expecting that mass vaccination campaigns will hopefully be performed in the next few months thanks to the development of different vaccines [132], pharmaceutical research may play a key role by implementing novel formulations for old active pharmaceutical compounds that are able to both enhance the delivery of the agents to the site of infection and improve safety by minimizing the occurrence of side effects, as seen not only for drugs but also for dietary supplements, herbal compounds, and medical gas mixtures.

## Figures and Tables

**Figure 1 pharmaceutics-13-00302-f001:**
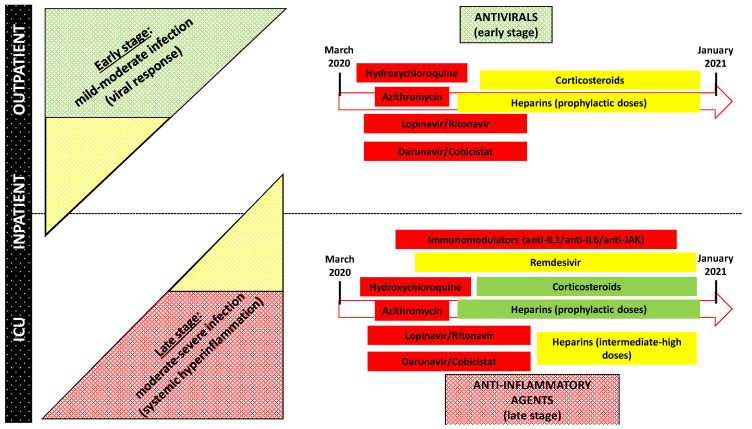
Relationship between severity of COVID-19, clinical scenarios, and use of specific repurposed COVID-19 agents. Green box: standard of care; Yellow box: recommended in selected cases; Red box: not recommended in clinical practice. ICU: intensive care unit. IL1: interleukin-1; IL6: interleukin-6; JAK: Janus kinase.

**Figure 2 pharmaceutics-13-00302-f002:**
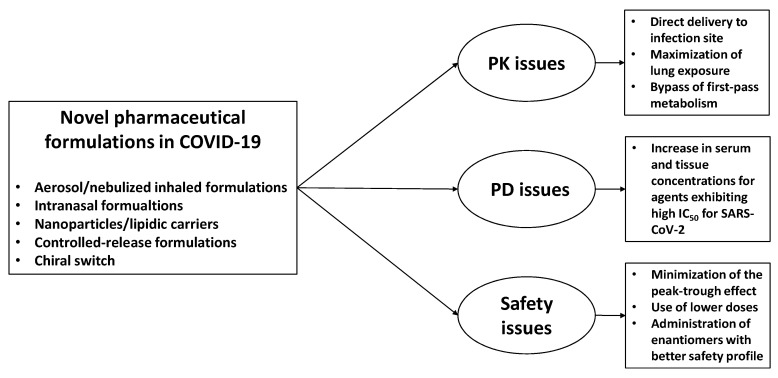
Relationship between pharmaceutical formulations and pharmacokinetic/pharmacodynamic properties affecting efficacy and safety of repurposed agents in respiratory viral diseases, including COVID-19 setting. PK, pharmacokinetic; PD, pharmacodynamic.

**Table 1 pharmaceutics-13-00302-t001:** Summary of the different approaches to drug repurposing potentially applicable to COVID-19. In bold, examples of repurposed drugs used for the management of COVID-19.

Repurposing Approach	Main Features	Examples of Repurposed Drugs
**Experimental approach**		
Binding assay	Identification of novel targets of known drugs through implementations of different techniques (e.g., mass spectrometry, affinity chromatography)	Imatinib for KIT-driven gastrointestinal stromal tumor
Phenotypic screening	Screening of compounds using in vitro or in vivo disease models	Clemastine and quetiapine for multiple sclerosis
**Computational approach**		
Signature matching	Comparation of the “signature” of a drug (such as its transcriptomic, structural, or adverse effect profile) with that of another drug or disease phenotype	Topiramate for inflammatory bowel diseases
In silico studies	Application of sophisticated analytical methods to existing data identifying novel potential associations between drug and disease—it can be classified in a molecular-based approach and real-world data-based approach	Chloroquine (91) * Hydroxychloroquine (262) * Pirfenidone (1) *
Pathway or network-based methods	Identification of repurposing targets by using network analysis based on genetic, protein, or disease data	Colchicine (24) * Melatonin (8) *
Drug centric	Association between known drugs and novel targets to predict new indications—it includes molecular docking (screening of single agents against a library of protein structures)	Mebendazole for inhibition of angiogenesis
Target based	Association between a known target and its established drug and a new indication—it requires a deep understanding of the molecular relationship between the target and the disease	
Retrospective clinical analysis	Systematic analysis of electronic health records, clinical trial data, and post-marketing surveillance data could provide information concerning drug repurposing	Aspirin for colorectal cancer
Artificial intelligence and machine learning models	Creation of a neural network based on the use of open chemical/drug database as input and the implementation of different algorithms to obtain the required drug as output	Baricitinib (15) *

* number of clinical trials according to *ClinicalTrials.gov* (search performed on 26 November 2020). KIT: tyrosine-protein kinase KIT CD117.

**Table 2 pharmaceutics-13-00302-t002:** Specific initiatives for acceleration of development support and evaluation procedures for COVID-19 treatments and vaccines implemented by the European Medicines Agency [34,35].

Specific Field	Description and Main Features
1. Rapid scientific advice	- No prespecified submission deadline - Flexibility regarding type and extent of the briefing dossier - Free of charge - Total review time reduced from regular 40–70 days to 20 days
2. Rapid agreement of a pediatric investigation plan and rapid compliance check	- No prespecified submission deadline - Possibility of a focused scientific documentation - Total evaluation time reduced to a minimum of 20 days compared to regular 120 days - Timeline for a compliance check can be reduced to 4 days
3. Rolling review	- an ad hoc procedure allowing EMA to continuously assess the data for an upcoming highly promising application as data become available, i.e., preceding the formal submission of a complete application for a new marketing authorization (or for an extension of indication in case of authorized medicines) - Several rolling review cycles can be carried out during the evaluation of one product as data continue to emerge, with each cycle requiring around two weeks
4. Marketing authorization	- Possibility of implementation of a rolling review - Possibility of application for accelerated assessment, with reduction of the review time from 210 days to less than 150 days
5. Extension of indication and extension of marketing authorization	- Rapid scientific advice, rapid agreement of a pediatric investigation plan, rolling review, and accelerated assessment can be implemented also for repurposed drugs
6. Compassionate use	- While coordination and implementation of a compassionate use program remain the competence of a Member State, EMA can provide through the Committee for Human Medicinal Products recommendations for a “group of patients” on a medicinal product eligible for the centralized procedure, in order to favor a common approach across Member States
7. Other considerations	- The Priority Medicines (PRIME) scheme could be considered by developers to receive enhanced support for the development of treatments or vaccines for COVID-19, thus facilitating accelerated assessment at the time of application for a marketing authorization - No specific considerations are provided regarding rapid reviews of orphan designations, given that applications for orphan designation in COVID-19 setting are not expected due to the high number of infections

**Table 3 pharmaceutics-13-00302-t003:** Summary of novel relevant pharmaceutical formulations for repurposed drugs implemented for COVID-19.

Repurposed Agents	Novel Formulations for Targeted Tissue Delivery	Galenic Formulations	Novel Formulations Aiming to Improve Safety	Novel Formulations with Adjuvant Micronutrients
Hydroxychloroquine	Inhaled formulations via nebulization (developed at clinical stage—phase 1/2) Liposomal formulation for intratracheal instillation (preclinical model)	Oral solution/suspension for nonresponsive patients or children (currently used in clinical settings)	Chirality switch with proposed use of (S)-(+)-enantiomer (working hypothesis)	Association with zinc supplement to improve inhibitory activity on SARS-CoV-2 replication (ongoing clinical trials) Formulation with glycyrrhizic acid to enhance delivery with synergic antiviral and anti-inflammatory effect
Remdesivir	Inhaled formulations via nebulization or dry powder (developed at clinical stage—phase 1/2)	-	-	-
Lopinavir/Ritonavir	Use of protease inhibitors encapsulated in endogenous/exogenous extracellular vesicles for targeted delivery (working hypothesis)	Use of protease inhibitors encapsulated in endogenous/exogenous extracellular vesicles for targeted delivery (working hypothesis)	Use of protease inhibitors encapsulated in endogenous/exogenous extracellular vesicles for targeted delivery (working hypothesis)	-
Heparin and fibrinolytics	Inhaled formulations of unfractionated heparin via nebulization (ongoing clinical trials in COVID-19 patients) Inhaled formulation of alteplase via nebulization has been developed (phase 2 trial in non-COVID patients)	-	-	-

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
