# Peer review of "Drug Repurposing in the COVID-19 Era: Insights from Case Studies Showing Pharmaceutical Peculiarities"

_pharmaceutics, 2021, doi:10.3390/pharmaceutics13030302_

Round 1

Reviewer 1 Report

This review article describes and gives comments about repurposing a number of medicines for treatment of COVID-19. It provides valuable references for future research regarding effective and safe treatments of COVID-19.

Minor comments:

Table 1: Chloroquine (91)* The reference #91 (line 880) is about PDE4 not chloroquine.

Line 248: “peak-through”, misspelled?

Reviewer 2 Report

This article is a narrative review that aims to describe pharmaceutical aspects of various case studies involving COVID-19 treatment using repurposed drugs, in most cases off-label compounds. It is a useful text summarizing the contemporary status of knowledge on still rather hopeless ways of COVID-19 treatment. The focus has been placed on pulmonary drug delivery. Therefore, novel approaches of drug delivery like nebulization, aerosols, inhaling various types of drug mists, intranasal applications, etc. are discussed at length, including technological aspects and physiochemical properties of compounds in question.

I am with the authors that there is a ‘tsunami’ of trials to repurpose older and newer drugs to increase the effectiveness of COVID-19 treatment. That ‘tsunami’ somehow translated into difficult to regurgitate information in the article as well. The surplus of information, not always necessary for the logical flow of the text, makes it difficult to localize the main thoughts of the authors, where they stand on the issue, and what they stand for. Perhaps, a narrative review does not require much of the own intellectual input, but it would be better for the reader to be directed at what message would be most worthwhile to take from the cacophony of information, at least as suggested by the authors’ expertise.

Specific comments and/or suggestions are as follows:

1/ reverse the structure of the text by moving ‘Drug repurposing in COVID-19: Case studies’ somewhere upfront from the end of the article, according to the purpose of your writing and also better from the clinical and practical viewpoint.

2/ focus on the pulmonary route of drug administration should be mitigated. This route is most suitable in the milder cases and, like remdesivir, is without any clear benefit in severe respirator-treated cases, the involvement of multi-organ systems, and does not change mortality, and is not devoid of extra complications either.

3/ you often write in very long entangled sentences, which is bad and difficult to comprehend expression of in-itself complex issues. Sentences of 5-7 lines in length, e.g., l. 178-196 should be broken into three or so; in other fragments of the text as well.

4/ l. 237 – wrong grammar (efficacy ??)

5/ you missed the anticancer plitidepsin, derived from a marine animal, currently top trendy and promising, 30x in vitro stronger acting against SARS-CoV-2 than remdesivir. Looks much more promising, at least for now, than many other tackled in the article.

Reviewer 3 Report

Well done review article with good amount of data. No additional comments

Reviewer 4 Report

Gatti and De Ponti in their manuscript reviewed the idea of drug repurposing to fight the SARS-CoV2 infection. The manuscript is very well written and it was my pleasure to read it. Therefore, I recommend to accept the paper in present form.

Reviewer 5 Report

The topic is of interest while the manuscript reads too lengthy. I would like to see a more succinct version. Figure 2 is not specific to COVID-19 but generally relevant to all respiratory viral diseases.
